



# Source Apportionment and Evolution of N-containing Aerosols at a Rural Cloud Forest in Taiwan by Isotope Analysis

Ting-Yu Chen[1], Chia-Li Chen[1], Yi-Chi Chen[2], Charles C.-K. Chou[3], Haojia Ren[*,2], and Hui-Ming Hung[*,1]

[1]Department of Atmospheric Sciences, National Taiwan University, Taipei, 10617 Taiwan
[2]Department of Geosciences, National Taiwan University, Taipei, 10617 Taiwan
[3]Research Center of Environmental Changes, Academia Sinica, Taipei, 11529 Taiwan

*Correspondence to*: Hui-Ming Hung (hmhung@ntu.edu.tw)

**Abstract.** Ammonium and nitrate are two major N-containing aerosol compositions. The deposition of N-containing aerosols
has impacts on regional ecology and the biogeochemical cycle. In this study, aerosols in a rural cloud forest (Xitou in Taiwan)
were studied using $^{15}N$ and $^{18}O$ isotope analysis to assess the sources and formation pathways of the local N-containing aerosols
linking to a metropolitan. Aerosol samples were collected for different size ranges using a micro-orifice uniform deposit
impactor (MOUDI) on a half-day basis in December 2018. The chemical functional groups were analyzed using a Fourier
transformed infrared spectroscopy with attenuated total reflection technique (FTIR-ATR), while the isotope analysis was
performed using a gas chromatography-isotope ratio mass spectrometer (GC-IRMS). The average measured aerosol
concentration ($PM_{10}$) was 0.98 (ranging from 0.15 to 3.31) and 0.25 (ranging from 0.00 to 1.51) µg/m$^3$ for $NH_4^+$ and $NO_3^-$,
respectively. In general, a higher functional group concentration was observed during the daytime by a factor of 1.5 to 6 than
nighttime, likely due to the transportation of pollutants from upper stream urban and industrial regions through the local sea
breeze combined with valley wind. The presence of fog can further elevate the concentration by a factor of 2 to 3, resulting
from the stronger inversion and lower boundary layer height. The higher $NH_4^+$ concentration in fine particles under foggy
conditions can further promote submicron-sized $NO_3^-$ formation via aqueous phase dissolution with $NH_4^+$ neutralization.
Furthermore, the higher RH during fog events shifted the mass distribution of aerosol functional groups to a larger mode size.
By comparing the $\delta^{15}N$ value directly or the analysis using a statistical isotope mixing model, MixSIAR, the $NH_4^+$ is probably
originated from the industries, coal-fired power plants (CFPP), or fertilizer plants, while $NO_3^-$ might be contributed from the
CFPP, industrial or urban sources. The overall $\delta^{18}O$ of $NO_3^-$ is +72.66‰ ± 3.42‰, similar to that in other winter Asia studies,
suggesting the major formation pathway via $O_3$ oxidation ($\delta^{18}O$ = +72.5 to 101.67‰). However, a lower $\delta^{18}O$ (< +67‰) for
particles less than 0.56 µm during foggy daytime suggests the local contribution via the peroxyl radical oxidation before
partitioning into aerosol phase under foggy conditions. Overall, the $\delta^{15}N$ and $\delta^{18}O$ distribution profiles as a function of particle
size in the studied rural forest site reveal the evolution of aerosol composition from remote coastal regions with chemical
processes along the transport process, which can be further affected by weather conditions such as fog events.

**Keywords:** aerosol, fog, functional group, nitrogen isotope, oxygen isotope



# 1 Introduction

Aerosols play an essential role in weather, climate, ecology, and human health (Poschl, 2005; Seinfeld and Pandis, 2006). Aerosols are mainly composed of sulfate, nitrate, ammonium, and other organic species. Nitrogen is one of the significant elements of aerosol in various forms, such as ammonium, nitrate, organic nitrogen, etc. Ammonium and nitrate are the primary N-containing alkaline and acid compositions, respectively, and the balance of the ions can influence aerosol acidity. Also, the local weather, such as fog formation, can be affected by the aerosol characteristics via the hygroscopicity of aerosol composition (Petters and Kreidenweis, 2007). Furthermore, the N-containing aerosols not only affect human health and climate but also play an important role in the regional and global nitrogen biogeochemical cycles. The remote transportation of N-containing aerosols from human activities may result in additional nutrient input at deposition sites, affecting local plant growth and ecology (Bobbink et al., 2010). Therefore, the amount of the N-containing aerosols formed and transported to the rural area and their potential sources should be investigated to evaluate the origin of the N-containing species and their impacts.

Ammonium in aerosols could form from gaseous ammonia, mainly generated from agricultural activities (Behera et al., 2013). Besides, $NH_3$ from fossil fuel exhaust and slipping during selective catalytic reduction (SCR) processes also contribute to aerosol $NH_4^+$ formation (Cape et al., 2004). Nitrate in aerosols is produced by oxidation of its precursors, nitrogen oxides ($NO_x$), emitted from fossil fuel combustion, biomass burning, lightening, and biogenic soil emission. The formation pathway of aerosol $NO_3^-$ varies with conditions. In the daytime, NO can be oxidized by $O_3$ or peroxyl radicals to form $NO_2$, which could be photolyzed back to NO or further react with OH radicals to generate nitric acid, forming the nitrate aerosols. At night, $NO_2$ may further be oxidized to $NO_3$, reacting with other $NO_2$ to form $N_2O_5$. The hydrolysis of $N_2O_5$ gives another pathway to form nitrate aerosols (Jacob, 1999; Seinfeld and Pandis, 2006).

The stable nitrogen isotope in aerosols provides a clue to distinguish the probable sources of nitrogen content. Since the abundance of $^{15}N$ and $^{14}N$ in gaseous precursors of $NH_4^+$ and $NO_3^-$ varies in different emission sources, the $\delta^{15}N$, defined as $((^{15}N/^{14}N)_{sample}/(^{15}N/^{14}N)_{air} - 1) \times 1000$ (‰), can act as an indicator of the associated nitrogen species (Felix et al., 2012; Felix et al., 2014; Walters et al., 2015; Pan et al., 2016; Chang et al., 2016; Savard et al., 2017; Pan et al., 2018a; Zhang et al., 2020). For nitrate, not only the $\delta^{15}N$ can be an index of sources, but the $\delta^{18}O$, defined as $((^{18}O/^{16}O)_{sample}/(^{18}O/^{16}O)_{VSMOW} - 1) \times 1000$ (‰), where VSMOW stands for Vienna Standard Mean Ocean Water, can reveal the oxidation pathway (Fig. 1) of nitrate formation due to the $\delta^{18}O$ difference between its oxidants: $O_3$, OH, $RO_2$ (including hydrogen peroxyl and organic peroxyl radicals), and $H_2O$ (Hastings et al., 2003; Fang et al., 2011; Gobel et al., 2013).

Xitou, an experimental forest of National Taiwan University, is a planted forest located in central Taiwan. As the origin of Beishih brook, Xitou is in the position of a river valley topography towards the northwest, connecting to Taichung City Metropolitan. Due to the topography, the sea breeze combined with mountain-valley wind dominates the diurnal local circulation, bringing air mass from different regions between daytime and nighttime. During the daytime, the sea breeze combined with valley wind can bring pollutants along the transporting path from coastal areas passing through the coal-fired power plants, industrial sites, and cities. As the wind direction reverses during nighttime, the pollutant concentration decreases



(Chen et al., 2021). Besides, the afternoon upslope fog occurs frequently in the Xitou forest due to the boundary layer inversion and the sea breeze combined with valley wind (Hsieh, 2019). Therefore, the fog might affect aqueous chemical processes locally.

The analysis of δ15N and δ18O for nitrogen-associated species as a function of particle size might provide the origin of the N-containing species and the evolution of transport and chemical processes. This study aims to investigate: 1) the interaction

between local circulation and the aerosol composition in a rural forest area linking to a city, 2) how the weather affects the aerosol composition in different sizes, and 3) source apportionment of rural N-containing aerosols by isotopic analysis.

## 2 Experiment Setup

A field campaign was conducted in Xitou experimental forest (23°40'12" N, 120°47'54'' E, 1,179 m a.s.l.) from 1st to 24th December 2018 to investigate the interaction between air quality, local circulation, and human activities in central Taiwan. To

dig into the link between local circulation and aerosol concentration and composition, daytime and nighttime aerosol samples in different sizes were collected using a cascade impactor, and underwent Fourier transformed infrared spectroscopy (FTIR) analysis for the functional group concentration (Coury and Dillner, 2008; Hung et al., 2016). Furthermore, δ15N and δ18O of N-containing species were measured using the denitrifier method (Sigman et al., 2001; Casciotti et al., 2002). The period mass-averaged δ15N values were further analyzed using a mixed stable isotope analysis in R (MixSIAR) model (Stock et al., 2018)

to resolve the potential sources of aerosol, while δ18O acts as an indicator of the oxidation process for nitrate formation in aerosols.

### 2.1 Sample collection

Ambient aerosol samples were collected using a 13-stage MOUDI (micro-orifice uniform deposit impactors, Model 125R, MSP Corporation, Shoreview, Minnesota, USA) with 46.2 mm polytetrafluoroethylene (PTFE) membrane filters (Whatman

7592-104). The cut-off size of MOUDI was 0.01, 0.018, 0.032, 0.056, 0.1, 0.18, 0.32, 0.56, 1.0, 1.8, 3.2, 5.6 and 10 μm, respectively, and the flow rate of sampling air was 10 L min$^{-1}$. The samples were categorized into daytime and nighttime to investigate the impact of diurnal mountain/valley-breeze circulation on aerosols. Daytime samples were collected from ~9:00 to ~17:00 (local time), and nighttime samples were from ~18:00 to ~6:00 the next day to represent the valley and mountain breeze, respectively. 20 sets of filter samples were collected from 2nd December 2018 to 22nd December 2018, including 4

foggy samples (181207D, 181213N, 181214D, 181215D, YYMMDD Daytime/Nighttime) and 16 non-foggy samples (181202D/N, 181207N, 181208D/N, 181209D/N, 181214N, 181215N, 181216D/N, 181220N, 181221D/N, 181222D/N). The collected filter samples were sealed, covered by aluminum foil, and preserved under 4℃ till the laboratory analysis to prevent contaminations.



## 2.2 FTIR-ATR Analysis

The concentrations of functional groups such as $NH_4^+$, $NO_3^-$ and $SO_4^{2-}$ were determined via FTIR measurement (Nicolet 6700, Thermo Fisher Scientific, USA) equipped with a single-reflectance attenuated total reflectance (ATR) monolithic diamond accessory (GladiATR™, PIKE Technologies, USA). Filter samples were pressure-pressed onto the ATR crystal to ensure a closed contact with the crystal. The infrared spectra were scanned at wavenumber from 4000 to 500 cm$^{-1}$ at a resolution of 1 cm$^{-1}$ as shown in Fig. S1. The selected spectrum for a given wavenumber range was fitted with one or multiple Lorentzian

curves to derive the peak absorbance (I) of each functional group. The curve fitting function can be written as:

$$A(\nu) = I \times \frac{\sigma^2}{4(\nu - \nu_{peak})^2 + \sigma^2} \qquad (1)$$

where $A(\nu)$ is the distribution of a specific absorption curve, $\nu$ is the wavenumber, and $\sigma$ is the scale parameter (half-width at half-maximum), which is the width of the absorption curve. For a mixture, the observed spectrum is a superposition of each substance i:

$$A\left(\nu, (\nu_{peak,1}, \sigma_1, I_1), (\nu_{peaki2}, \sigma_2, I_2), \ldots \right) = \sum_i A_i(\nu) = \sum_i I_i \times \frac{\sigma_i^2}{4(\nu - \nu_{peak,i})^2 + \sigma_i^2} \qquad (2)$$

The analyzing peak was ~1350 cm$^{-1}$ for nitrate and ~1417 cm$^{-1}$ for ammonium (Fig. S2); besides, the absorbance peak at ~1080 cm$^{-1}$ for $SO_4^{2-}$ was applied in a 3-curve fitting because of the nearby absorbance of the PTFE filters (Fig. S3). Therefore, the calibration of absorbance to concentration was based on the former analysis using the correlation of absorbance of FT-IR functional groups to the water-soluble ions measured by ion chromatography (Huang, 2016).

## 2.3 Isotope Analysis

### 2.3.1 Sample Analysis

Due to the instrumental detection limit, 10 sets of aerosol samples with higher N-containing functional group concentration under distinct weather conditions were selected for $\delta^{15}N$ and $\delta^{18}O$ isotope analysis of N-containing species (181202D/N, 181213N, 181214D/N, 181215D, 181220N, 181221D, 181222D, 181222N). Because the isotope analysis requires at least five

nmol of equivalent N in less than 5mL solution (i.e., the molar concentration of $NO_3 + NH_4 \geq 1\mu M$ N), the FTIR measurements provide a quantitative reference to infer the concentration of dissolved N-containing species. If the predicted concentration of one filter was too low, 2 to 4 filters collected on the same day with adjacent size bins were put together in a bottle during the rinsing process to ensure sufficient sensitivity for isotope analysis. Filter samples were cut in half and soaked into 30 mL Milli-Q water (resistivity = 18.2 MΩ at 25 ℃) and underwent a 30-minute ultrasonication to dissolve the water-soluble ions into the

solution. Afterward, the extracted solution was filtered through a 0.22 μm Millipore syringe filter and then preserved in an HDPE bottle. The samples were analyzed for the $\delta^{15}N$ of total nitrogen (TN) and nitrate + nitrite (NN), and the $\delta^{18}O$ of NN by the bacterial "denitrifier method" as stated by Sigman et al. (2001), Casciotti et al. (2002), and updated by Weigand et al. (2016). For TN analysis, the oxidation process by adding potassium persulfate in NaOH solution was to oxidize $NH_4^+$ and





other N-containing species in a reduced state into $NO_3^-$ before bacterial digestion. The isotope $^{15}N/^{14}N$ and $^{18}O/^{16}O$ was

measured using a gas chromatography-isotope ratio mass spectrometer (GC-IRMS) composed of a GC column system coupled

with Thermo MAT 253 Plus 10 kV IRMS. The international standard IAEA NO3 ($\delta^{15}N$ = 4.7 ‰, $\delta^{18}O$ = +25.61‰) and USGS

34 ($\delta^{15}N$ = -1.8 ‰, $\delta^{18}O$ = -27.93 ‰) were applied for $\delta^{15}N$ and $\delta^{18}O$ calibration (Bohlke et al., 2003). In each batch of

measurement, three to five duplicates of standards and bacteria blank were used to ensure the efficiency of bacterial conversion

and the stability of mass spectroscopy.

130    Ammonium is a major N-containing component in aerosols as part of TN. Since the concentration of water-soluble TN minus

NN correlates well with the measured $NH_4^+$ concentration from FTIR (Fig. S4), the water-soluble TN-NN can seem as $NH_4^+$.

Therefore, the $\delta^{15}N$ of ammonium can be derived by assuming the collected aerosol mainly comprised of nitrate, nitrite, and

ammonium with negligible other N forms such as organic nitrogen. The $\delta^{15}N$ of $NH_4^+$ can be calculated using Eq. (3) as

follows:

135    $$\delta^{15}N_{NH_4^+} = \frac{\delta^{15}N_{TN} \times M_{TN} - \delta^{15}N_{NN} \times M_{NN}}{M_{TN} - M_{NN}} \qquad (3)$$

where $M_{TN}$ and $M_{NN}$ are the molarities of total nitrogen (TN) and nitrate plus nitrite (NN) of the sample solution, respectively.

Additionally, since the aerosol nitrite concentration is mostly negligible based on ion-chromatography (IC) analysis of $PM_{10}$,

NN is assumed to be in $NO_3^-$ form, i.e., $\delta^{15}N$ of p-$NO_3^- \approx \delta^{15}N_{NN}$.

**2.3.2 Baysian Mixing Model Application**

140    A Bayesian mixing model, MixSIAR, was applied to assess the contribution of multiple aerosol sources. MixSIAR is a

statistical model used to infer the probability of mixture sources by analyzing their tracer composition, such as stable isotope

or fatty acids (Stock et al., 2018). In this study, the mass-weighted $\delta^{15}N$ of $NH_4^+$ and $NO_3^-$ at each sampling period was used

as the observation data. For simplification, the source data adopted the results of Savard et al. (2017) as summarized in Table

1 by assuming that the $\delta^{15}N$ of $NH_4^+$ and $NO_3^-$ was directly related to their emission sources, either single source or mixture

145    from those sources. The source data include $\delta^{15}N$ values from traffic, chemical and metal industries, feedlots, fertilizer plants,

and coal-fired power plants (CFPP) for $NH_4^+$, and $\delta^{15}N$ values from traffic, chemical and metal industries, fertilizer plants and

oil refinery, and CFPP for $NO_3^-$ nitrogen source apportionment. The gas compressors source was not considered in this study.

**3 Results and Discussion**

**3.1 Functional group concentration by FTIR-ATR**

150    The averaged functional group concentration measured using FTIR-ATR of collected 0.01 to 10 μm samples was $NH_4^+$: 0.98,

$NO_3^-$: 0.25, $SO_4^{2-}$: 5.16, and black carbon (BC): 0.81 with the unit of μg/m$^3$ as summarized in Table 1. The mass concentration

distribution of $NH_4^+$ and $NO_3^-$ as a function of aerosol size is shown in Fig. 2. $NH_4^+$ is mainly distributed in submicron mode,

with the most significant mass concentration in 0.32-0.56 μm. The $NO_3^-$ during the non-foggy period mostly appears in sizes

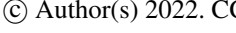


larger than 1 μm and peaks at 3.2-5.6 μm. The mass distribution pattern of $SO_4^{2-}$ mainly in the sub-micron mode is consistent
with that of $NH_4^+$ (Fig. S5), which suggests that most ammonium is in the form of sulfate-associated salts. On the contrary,
$NO_3^-$ in the aerosol is mainly formed from the substitution reaction of sea salt aerosol or dust in the larger size (> 1 μm) aerosols
by $HNO_3$ (Evans et al., 2004). The non-observed nitrate in submicron particles during non-foggy days is likely due to the
thermodynamic equilibrium under ammonia-limited conditions (Seinfeld and Pandis, 2006). Generally, the functional group
concentration was higher for the daytime than that at night (Table 1). Foggy weather also promoted a higher concentration.
The influence of weather on the mass concentration distribution is discussed in the following subsections.

### 3.1.1 Difference between daytime and nighttime

The functional group concentration of $NH_4^+$ (1.00 μg/m$^3$) and $NO_3^-$ (0.25 μg/m$^3$) during non-foggy daytime was higher than
that in non-foggy nighttime (0.56 μg/m$^3$ and 0.04 μg/m$^3$, respectively) as shown in Table 1, and $SO_4^{2-}$ and BC also have
approximately 1.5 times higher concentration during non-foggy daytime. The greater daytime concentration might link to the
upstream transportation of urban pollutants by valley wind combining with the sea breeze (Chen et al., 2021). Once the wind
direction changes into mountain wind accompanying land breeze, the cleaner upper-stream air diluted the pollutants in the
Xitou Forest area.

### 3.1.2 The Influence of Fog

The daytime concentration of $NH_4^+$ and $NO_3^-$ was 2 to 4 times higher in the foggy period than that in the non-foggy period
(Table 1), and the mass distribution seems to shift to a larger size mode for $NH_4^+$ as shown in Figs. 2(c) and 2(d). Higher
ammonium nitrate concentration might result from the stronger boundary layer inversion on foggy days. When the boundary
inversion gets stronger in Xitou area, the moisture transportation by upwelling turbulence is weakened. Therefore, water vapor
could accumulate in the lower atmosphere, promoting fog formation and prolonging fog lifetime (Hsieh, 2019). Furthermore,
the weakened upward transport could also accumulate pollutants in the lower boundary layer, causing a higher concentration
in the ambient atmosphere. The concentration of black carbon (BC) also increased during foggy periods, as shown in Table 1
and Fig. S5. This might reveal the inference of the boundary layer on the higher aerosol concentration because BC is a primary
aerosol with limited chemical reactions in the atmosphere.

The mass distribution of $NH_4^+$ shifted slightly to a larger size mode on foggy days. According to a calculation of simultaneously
observed dry and wet aerosol size distribution in Xitou, $NH_4^+$-containing aerosol has a hygroscopicity coefficient κ = 0.21±0.01
(Chen et al., 2021). The hygroscopic growth of aerosol from averaged RH of 80% under non-foggy circumstances to over 99%
during the foggy period could lead to a larger wet aerosol size. Extra high $NO_3^-$ concentration of 0.56-1 μm aerosol was
observed during foggy periods accompanied with the high $NH_4^+$ concentration in that size bin (Fig. 2(d)). In foggy periods,
the higher water content of aerosol promotes an aqueous phase reaction of aerosol $HNO_3$ uptake, and the higher concentration
of $NH_4^+$, more than $2×[SO_4^{2-}]$, gives extra neutralizing ion to stabilize the $NO_3^-$ as suggested by Chen et al. (2021).



### 3.2 Isotope Analysis of N-containing species

The $\delta^{15}N$ of $NH_4^+$ and $NO_3^-$ discussed in this section infers the probable aerosol sources, and the photo-oxidation processes of $NO_x$ in the atmosphere are inferred using measured $\delta^{18}O$ of $NO_3^-$. The isotope value of each sample is shown in Fig. 3, and the period mass-weighted averaged $\delta^{15}N$ and $\delta^{18}O$ are summarized in Table S1.

### 3.2.1 $\delta^{15}N$ of $NH_4^+$

Figure 3(a) shows the $\delta^{15}N$ value of aerosol $NH_4^+$ as a function of geometric averaged particle diameter. The $\delta^{15}N$ varies from -3.70‰ to +21.39‰, and the average mass-weighted $\delta^{15}N$ value is +11.95‰ with a standard deviation of 2.65‰. The $\delta^{15}N$ of 0.32-1 μm aerosols is in the range of +7.16‰ to +18.64‰, relatively higher than that of other size bins. The trend of a higher $NH_4^+$ $\delta^{15}N$ in submicron aerosols was also observed in Beijing (Pan et al., 2016; Pan et al., 2018b) but was approximately 12‰ lower in general. This offset probably results from the different emission sources or the partitioning processes. Overall, the processes forming aerosol $NH_4^+$ may lead to the size differentiated $\delta^{15}N$.

The daytime $\delta^{15}N$ of $NH_4^+$ is mostly greater than the nighttime one as summarized in Table 3, likely resulting from the different sources, such as transportation of high-$\delta^{15}N$ $NH_3$ from urban rush-hours traffic or industrial sources by sea breeze combined with the valley wind. As the mountain wind dominates after sunset, $NH_3$ might be attributed to the daytime residual (having lower $\delta^{15}N$ due to the daytime fractionation) or the local biogenic sources having a lower $\delta^{15}N$.

Fog varies the composition mass distribution among different size bins and can affect the isotopic ratio. Under foggy conditions, especially in foggy daytime, the $\delta^{15}N$ value of larger size aerosols ($PM_{1-10}$-$NH_4^+$) was higher than non-foggy days and could be up to 21.39‰, similar to that of 0.32-1μm aerosols. As stated in section 3.1, $NH_3$ in higher concentrations can promote the partition of $HNO_3$ during foggy conditions. The larger $\delta^{15}N$ might result from the hygroscopic particle growth of $PM_1$-$NH_4^+$ with high $\delta^{15}N$. As $NH_4^+$ is likely to deliquesce to the liquid phase under high RH conditions, the gas-liquid phase transition could accompany isotope equilibrium fractionation for most aqueous particles (Walters et al., 2018). The $NH_3$-rich and high RH conditions might cause the $NH_3$ partition to condensed phase favors higher $\delta^{15}N$ during equilibrium fractionation processes (Pan et al., 2018b). On non-foggy days, having a relatively lower composition concentration with more acidic properties (indicating $NH_3$ limited), a higher portion of $NH_3$ might participate in the aerosol phase to lead a lower $\delta^{15}N$-$NH_4^+$ toward the original $\delta^{15}N$-$NH_3$.

### 3.2.2 $\delta^{15}N$ of $NO_3^-$

The $\delta^{15}N$ value of $NO_3^-$ as a function of size bin shown in Fig. 3(b) ranges from -1.07 to +6.64‰, with a mass-weighted mean value of +2.98‰ and a standard deviation of 1.20‰. This value agrees with other studies measured in Asia or the Pacific Ocean in winter to spring period (-1‰ ± 3‰ in spring by Guha et al. (2017); 2.0‰ ± 0.4‰ in spring and 8.6‰ ± 0.4‰ in winter by Kim et al. (2019); 3.1 ± 1.1 ‰ in winter by Kawashima (2019)). Since the mass distribution of $NO_3^-$ different from $NH_4^+$ and varies under foggy and non-foggy conditions (Fig. 2), the discussion of $NO_3^-$ $\delta^{15}N$ is then focused on a large mode





$PM_{1-10}$-$NO_3^-$ (1μm < geometric diameter < 10 μm) and a small mode $PM_1$-$NO_3^-$ (< 1μm) particles separately. $PM_{1-10}$-$NO_3^-$ is available for most samples, but $PM_1$-$NO_3^-$ is limited to foggy daytime due to the available alkaline species as stated in section 3.1.2. For a given sampling period, $PM_1$-$NO_3^-$ has lower $\delta^{15}N$ (-1.07 to +3.19‰) than $PM_{1-10}$-$NO_3^-$ (+1.85 and +6.64‰), likely due to different formation processes. $PM_{1-10}$-$NO_3^-$ might be formed through the reaction of $HNO_3$ or $NO_2$ with the coarse

particles composing NaCl or dust (Evans et al., 2004; Hoffman et al., 2004) during the transport from the coast through the urban region and further to Xitou. Therefore, a higher $\delta^{15}N$ $NO_3^-$ participates in the aerosol-phase through isotopic equilibrium fractionation with lower $\delta^{15}N$ $HNO_{3(g)}$ or $NO_2$ gas molecules remaining in the air (Walters and Michalski, 2015). In contrast, $PM_1$-$NO_3^-$ only occurs in the foggy days, likely forming in the mountain region with high water content and available $NH_3$. The available $HNO_{3(g)}$ for $PM_1$ is from the residual $NO_x$ (after reacting with coarse mode particles at the upper stream) and has

lower $\delta^{15}N$ compared to $PM_{1-10}$-$NO_3^-$. The $PM_1$-$NO_3^-$ formed through the aqueous phase reaction under high $NH_4^+$ with effective gas-phase $NO_2$ uptake might have a limited isotopic selection which leads to a low $\delta^{15}N$ of $NO_3^-$ under foggy conditions.

The sample of 21D is a special case with higher $\delta^{15}N$ values. It might result from the agricultural activities nearby, including fertilizing and mowing. The fertilizer generates $NO_x$ with higher $\delta^{15}N$ (Savard et al., 2017), which indicates that the agricultural

activities might cause higher $\delta^{15}N$ values than other days.

### 3.2.3 $\delta^{18}O$ of $NO_3^-$

The $\delta^{18}O$ of $NO_3^-$ ranged from +53.90 to +79.81‰ (Fig. 3(c)), with a half-day period mass-weighted average of +72.66‰ and a standard deviation of 3.42‰. The results are within the $\delta^{18}O$ range observed in cool seasons over the Mt. Lulin site in Taiwan (69‰ ± 15‰ reported by Guha et al. (2017)) and also in the typical range of other studies (averaged value from 70.9‰ to 83.8

‰) (Savarino et al., 2007; Wankel et al., 2010; Fan et al., 2020; Sun et al., 2020). The relatively high $\delta^{18}O$ compared to summer samples (32 ± 13‰ reported by Guha et al. (2017)) indicates that most $NO_3^-$ precursors (i.e., $NO_x$) were formed by $O_3$ oxidation whether it was further oxidized through OH oxidation of $NO_2$ or $N_2O_5$ hydrolysis pathways (from +72.5‰ to +101.67‰, detailed in SI description). The slightly lower daytime $\delta^{18}O$ (+69.67‰ to +72.52‰ based on half-day average) compared to nighttime samples (+74.82‰ to +79.81‰) as shown in Table S1 indicates that peroxyl radicals might partially participate in

the daytime photooxidation processes or relatively lower $\delta^{18}O$ of OH leading to a lower $\delta^{18}O$ in nitrate aerosols during daytime as stated in other studies (Gobel et al., 2013; Hastings et al., 2003; Fang et al., 2011).

For $PM_1$, the $\delta^{18}O$ of 0.32-0.56 μm $NO_3^-$ under foggy conditions (+53.90‰ and +66.13‰ for 14th December and 15th December daytime sample, respectively) is relatively lower than that over larger sizes (e.g., +75.65‰ and +73.98 ‰ of 0.56-1 μm) suggesting the formation pathway difference. The concentration of 0.32-0.56 μm $NO_3^-$ is relatively lower than that of 0.56-1

μm or $PM_{1-10}$, and it might attribute to ambient air mass nearby the observation site. Because the fine particles are more acidic (Chen et al., 2021), $NO_3^-$ can frequently exchange with gas-phase $HNO_3$ to reveal the local $\delta^{18}O$ of $NO_3^-$. The peroxyl radicals derived from the biogenic volatile organic compounds photooxidation at Xitou forest area might be active oxidants locally for fine mode organic nitrate ($RONO_2$ or $ROONO_2$) and $HNO_3$ from $NO + RO_2 \rightarrow NO_2 + RO$ and $NO_2 + OH \rightarrow HNO_3$ oxidation


to have a lower $\delta^{18}O$ of $NO_3^-$ (SI description). On the other hand, the higher $\delta^{18}O$ of 0.56-1 μm $NO_3^-$ is likely formed from the growth of smaller particles and aqueous phase reactions such as $HNO_3$ partition, which could be neutralized by excess $NH_4^+$ at an earlier stage to be less influenced by peroxyl radicals. Furthermore, the $PM_{1-10}$-$NO_3^-$ are mainly produced nearby the urban regions via the reactions of $HNO_3$ or $NO_x$ with sea salt, i.e., $HNO_3 + NaCl_{(p)} \rightarrow HCl_{(g)} + NaNO_{3(p)}$ or $2NO_2 + NaX_{(p)} \rightarrow XNO_{(g)} + NaNO_{3(p)}$ (X = Cl or Br), which may also produce $NO_3^-$ with a higher $\delta^{18}O$ because most O atoms of $NO_3^-$ can come from $O_3$ during the fast $NO \leftrightarrow NO_2$ conversion processes (Gobel et al., 2013).

### 3.3 Source apportionment by isotope analysis

The $\delta^{15}N$ of collected $NH_4^+$ and $NO_3^-$ is applied for source apportionment since $\delta^{15}N$ in N-containing aerosol is dependent on the precursor sources (Felix and Elliott, 2014; Walters et al., 2015; Chang et al., 2016; Pan et al., 2016; Savard et al., 2017; Pan et al., 2018b; Fan et al., 2019). Figure 4 shows the averaged $\delta^{15}N$ under distinct weather conditions and the isotope value of single-source based on the observation by Sarvard et al. (2017). By assuming that the mass-weighted average isotope

represents the possible source contribution with a single source having similar $\delta^{15}N$ as reported by Sarvard et al. (2017) for simplification, the probable aerosol-N sources are summarized in Table 3. $NH_4^+$ might be originated from CFPP, traffic, or industries and least likely from feedlots. The urban sources or CFPP might contribute to $PM_{1-10}$-$NO_3^-$, and industries to the lower $\delta^{15}N$ of $PM_1$-$NO_3^-$ under foggy conditions. In contrast, the significant difference of $\delta^{15}N$ between measurement and fertilizer plants (+10.8‰) suggests the limited contribution of fertilizer production-related $NO_3^-$. Overall, the probable sources

of $NH_4^+$ and $NO_3^-$ were anthropogenically originated, such as CFPP, industries, and urban traffic. The sea breeze could transport the precursor gases or aerosol phase pollutants from coastal coal-fired power plants, industrial sources, or urban emissions to the forest area by upslope wind (Chen et al., 2021). During the transportation, the chemical reactions might further promote PM formation, having the measured $\delta^{15}N$ of collected samples close to that of the available gas-phase species.

As PM is a mixture attributed from various sources, the weight-averaged half-day $\delta^{15}N$ of $NH_4^+$ and $NO_3^-$ was analyzed using

the MixSIAR model to distinguish the posterior probability of aerosol sources as summarized in Table 4. The daytime samples of $21^{st}$ December were excluded in this analysis due to the interference from the agricultural activities nearby. With taking account of all weather conditions, the order of the possible sources from the highest to the lowest probability is {industries, CFPP, fertilizers, traffic, feedlots}. The first two sources have a higher likelihood, > 20%. As the conditions were divided with different weather patterns, fertilizer plants have increased the importance, especially for foggy daytime. Feedlots remain the

lowest. The model results agree with the direct comparison, indicating that the anthropogenic sources contribute significantly to aerosol-$NH_4^+$. The larger $\delta^{15}N$ during foggy daytime suggests a higher probability from fertilizer production, indicating the likelihood of locally produced ammonium from the fertilizer manufacturers nearby the agricultural area because of the lower wind speed and lower boundary layer height.

Though the $\delta^{15}N$ of $NO_3^-$ might seem similar and have no difference between each other, some trends could be revealed from

the MixSIAR model analysis. The MixSIAR results show that industries, urban, and CFPP are the major sources for both $PM_1$-$NO_3^-$ and $PM_{1-10}$-$NO_3^-$, whereas fertilizer plants have the lowest probability. The posterior probability of $PM_1$ and $PM_{1-10}$





nitrate sources has difference slightly: the $PM_{1-10}$ $NO_3^-$ was more likely from CFPP or urban sources, while industries took the majority of $PM_1$-$NO_3^-$ formation. The inferred source difference might suggest that the coarse mode aerosols came from the coastal sea salt particles mixing with the emission of coal-fired power plants or the Taichung-Changhua metropolitan during the inland transportation. On the other hand, $PM_1$-$NO_3^-$ is likely formed locally and might have a higher portion of nearby sources. For both $PM_1$ and $PM_{1-10}$ nitrate, fertilizer industry was the minority of the $NO_3^-$ sources in the Xitou forest area in this study, which is different from the result of $NH_4^+$. The discrepancy might result from the type of produced nitrogen fertilizers in the nearby area or the higher contribution of $NO_3^-$ from the power plant or urban sources through sea breeze and valley wind transport.

## 4 Conclusions

The mass distribution of aerosol $NH_4^+$ and $NO_3^-$ concentration and the associated isotope analysis were analyzed to investigate the evolution of nitrogen species before reaching the studied site. In Xitou forest, the average concentration of aerosol composition is 0.98 μg/m$^3$ for $NH_4^+$ and 0.25 μg/m$^3$ for $NO_3^-$. The 1.5 to 6 times higher concentration of $NH_4^+$ and $NO_3^-$ in the daytime indicates the local circulation combining land-sea breeze with mountain-valley wind could bring urban and industrial pollutants into the Xitou forests, further proved by the $\delta^{15}N$ analysis. The $\delta^{15}N$ of $NH_4^+$ from -3.70‰ to +21.39‰ with higher $NH_4^+$ $\delta^{15}N$ values of the 0.32-1 μm aerosols, where a higher concentration was measured, indicates that the aerosol was probably from the anthropogenic contribution by directly comparing with other studies or using the MixSIAR model. The $\delta^{15}N$ of $NO_3^-$ was from -1.07 to +6.64‰, with a mean value of 2.98‰ and a standard deviation of 1.20‰. Though the similar range of $NO_3^-$ among sources made it difficult to distinguish the origin of $NO_3^-$ directly, the statistical model still provided some hints: Industries, urban, and CFPP are the significant sources of particulate $NO_3^-$. The stronger boundary layer inversion during foggy days led to weaker upward transportation of air mass, causing a 2-3 times higher aerosol concentration. The mass distribution difference and the discrepancy of $\delta^{15}N$ of $NO_3^-$ between foggy and non-foggy conditions suggest that the additional $PM_1$-$NO_3^-$ for foggy days was formed locally with excess $NH_3$ in the aqueous phase. The difference of analyzed nitrogen sources between $PM_{1-10}$ and $PM_1$ $NO_3^-$ revealed the impacts of fog on aerosol formation: $PM_{1-10}$ was more likely produced by CFPP and urban area, whereas $PM_1$, only existed in the foggy period, had more local contributors such as a higher portion of industries. The inferred source difference might suggest that the N atoms of coarse mode aerosols came from the coastal sea salt particles mixing with the emission of coal-fired power plants or the Taichung-Changhua metropolitan during the inland transportation. On the other hand, $PM_1$-$NO_3^-$ is likely formed locally and might have a higher portion of nitrogen from nearby sources. However, the fractionation during the aerosol transportation under higher RH and high gaseous precursors can enlarge the isotope value in aerosol phases (Chang et al., 2018), which might affect the source apportionment results and should be appropriately assessed in the future. The observed $\delta^{18}O$ of $NO_3^-$ in this study, consistent with former studies conducted in a similar season, suggests that $O_3$ is the primary oxidant for $NO_x$ as a precursor of $NO_3^-$. The lower $\delta^{18}O$ value at 0.32-0.56 μm $NO_3^-$ under foggy daytime conditions indicates the participation of locally produced $RO_2$ in $NO_3^-$ formation. Overall, the



measured composition combined with the weather observation indicates the effects of local circulation and boundary layer on

air quality, and the isotope analysis further proved the influence of the inland transport from anthropogenic sources.

**Author contributions**

TY Chen and CL Chen carried out the field studies and aerosol composition analysis. TY Chen performed data analysis and MixSIAR model for N source apportionment and prepared the manuscript draft and editing. YC Chen and H Ren developed and conducted the isotope analysis. CCK Chou provides MOUDI instrumentation support and IC analysis of $PM_{2.5}$ and $PM_{10}$.

HM Hung supervised the project, including data discussion and manuscript editing.

**Competing interests**

The authors declare that they have no conflict of interest.

**Acknowledgments**

This study is supported by the Ministry of Science and Technology, Taiwan (108-2111-M-002-003, 109-2111-M-002-003,

and 110-2111-M-002-010) and National Taiwan University (110L892001). We acknowledge the local site support from the Administration of the Xitou Experimental Forest, College of Bio-Resources and Agriculture at National Taiwan University.

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



**Tables**

**Table 1. IR measured functional group average concentration of collected $PM_{10}$ under different weather conditions. (mean, [min, max] at the unit of $\mu g/m^3$)**

|  | Overall | non-foggy daytime | foggy daytime | non-foggy nighttime | foggy nighttime |
|---|---|---|---|---|---|
| **$NH_4^+$** | 0.98, [0.15, 3.31] | 1.00 | 2.48 | 0.56 | 1.12 |
| **$NO_3^-$** | 0.25, [0.00, 1.51] | 0.25 | 0.92 | 0.04 | 0.34 |
| **$SO_4^{2-}$** | 5.16, [0.62, 12.97] | 5.62 | 10.14 | 3.58 | 5.01 |
| **BC** | 0.81, [0.48, 1.46] | 0.95 | 1.25 | 0.59 | 0.71 |



**Table 2. Aerosol $\delta^{15}$N values of different sources used in this study. (Savard et al., 2017)**

| NH$_4^+$ source | NH$_4^+$ $\delta^{15}$N (mean ± SD) | NO$_3^-$ source | NO$_3^-$ $\delta^{15}$N (mean ± SD) |
|---|---|---|---|
| **CFPP** | 3.4 ± 10.4 | **CFPP** | 6.1 ± 2.0 |
| **traffic** | 17.1 ± 9.1 | **urban** | 5.7 ± 2.0 |
| **chemical and metal industries** | 11.0 ± 2.4 | **chemical and metal industries** | 1.0 ± 4.7 |
| **fertilizers plus oil** | 16.3 | **fertilizers plus oil** | 10.8 |
| **feedlots** | 27.7 ± 7.0 | | |





**Table 3. Mass-weighted isotope value (‰) and probable single source under distinct weather circumstances.**

|  | Non-foggy daytime | Foggy daytime | Non-foggy nighttime | Foggy nighttime |
|---|---|---|---|---|
| $\delta^{15}N$ of $NH_4^+$ (probable sources) | 13.20 (CFPP, traffic, industries) | 15.52 (traffic) | 9.30 (CFPP, traffic, industries) | 13.33 (CFPP, traffic, industries) |
| $\delta^{15}N$ of $PM_1$-$NO_3^-$ (probable sources) | - | 1.70 (industries) | - | 1.46 (industries) |
| $\delta^{15}N$ of $PM_{1-10}$-$NO_3^-$ (probable sources) | 2.72 (industries) | 3.98 (urban, industries) | 1.85 (industries) | - |
| $\delta^{18}O$ of $PM_1$-$NO_3^-$ | - | 70.48 | - | 79.81 |
| $\delta^{18}O$ of $PM_{1-10}$-$NO_3^-$ | 70.05 | 71.62 | 74.82 | - |




**Table 4. The posterior probabilities of aerosol sources inferred by MixSIAR (starred for the mean posterior probability greater than 20%.)**

| Weather condition (sample size, n) | NH$_4^+$ sources and posterior probabilities (Mean ± SD, %) | | | | |
|---|---|---|---|---|---|
| | CFPP | industries | feedlots | fertilizers | traffic |
| all cases (10) | 25.7 ± 15.1* | 32.5 ± 22.0* | 9.2 ± 8.2 | 17.7 ± 14.3 | 15.0 ± 13.5 |
| non-foggy day (3) | 19.0 ± 14.6 | 28.5 ± 20.0* | 13.8 ± 12.0 | 22.1 ± 17.4* | 16.6 ± 14.1 |
| foggy day (2) | 13.9 ± 12.5 | 24.2 ± 17.9* | 17.0 ± 13.6 | 27.2 ± 20.9* | 17.7 ± 14.8 |
| non-foggy night (4) | 21.0 ± 14.7* | 32.3 ± 21.2* | 10.6 ± 9.8 | 20.5 ± 16.4* | 15.5 ± 13.4 |
| foggy night (1) | 19.1 ± 15.0 | 23.1 ± 17.9* | 17.3 ± 14.3 | 20.5 ± 17.0* | 20.0 ± 15.9* |
| | PM$_{1-10}$-NO$_3^-$ sources | | | | |
| | CFPP | industries | fertilizers | urban | |
| all cases (5) | 27.2 ± 19.3* | 30.7 ± 17.8* | 13.9 ± 12.2 | 28.2 ± 19.8* | |
| non-foggy day (2) | 27.8 ± 19.7* | 25.2 ± 18.0* | 19.7 ± 15.6 | 27.4 ± 20.3* | |
| foggy day (2) | 28.0 ± 19.9* | 25.3 ± 17.3* | 19.2 ± 15.4 | 27.6 ± 19.4* | |
| non-foggy night (1) | 26.5 ± 20.5* | 27.2 ± 19.6* | 19.8 ± 16.2 | 26.5 ± 20.0* | |
| | PM$_1$-NO$_3^-$ sources | | | | |
| | CFPP | industries | fertilizers | urban | |
| all cases (3) | 23.8 ± 18.2* | 36.5 ± 21.0* | 14.6 ± 13.3 | 25.1 ± 19.2* | |
| foggy day (2) | 26.6 ± 19.9* | 30.0 ± 19.4* | 16.6 ± 14.8 | 26.7 ± 19.7* | |
| foggy night (1) | 27.4 ± 19.9* | 26.9 ± 19.3* | 19.1 ± 15.8 | 26.6 ± 19.7* | |





**Figures**

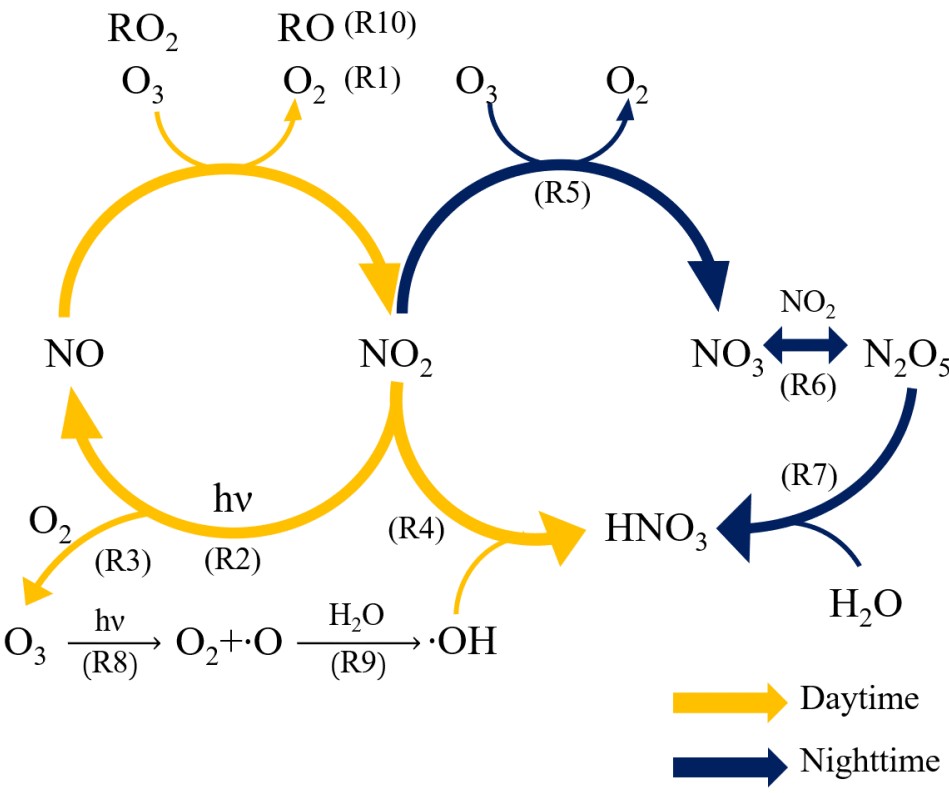


**Figure 1: The formation pathway of nitric acid to form aerosol nitrate during daytime and nighttime.**

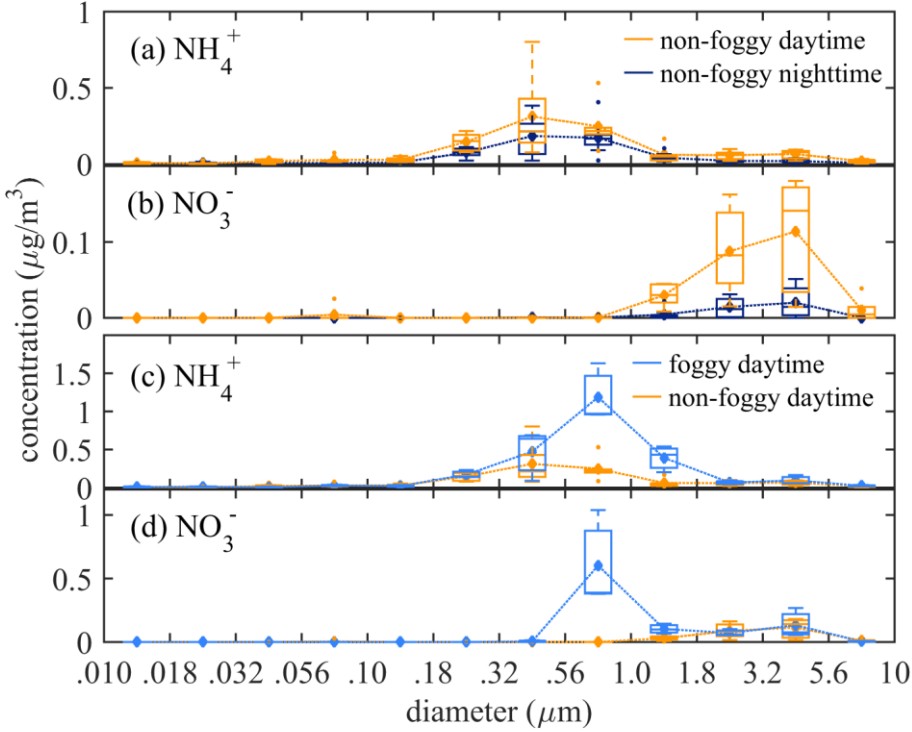

**Figure 2: The statistical box plot of concentrations as a function of size bin at non-foggy daytime and nighttime for (a) NH₄⁺, and (b) NO₃⁻, and at foggy and non-foggy daytime for (c) NH₄⁺ and (d) NO₃⁻. (diamond: mean value; outliers: < 1st quartile Q1-1.5 interquartile range (IQR) or > 3rd quartile Q3+1.5 IQR).**


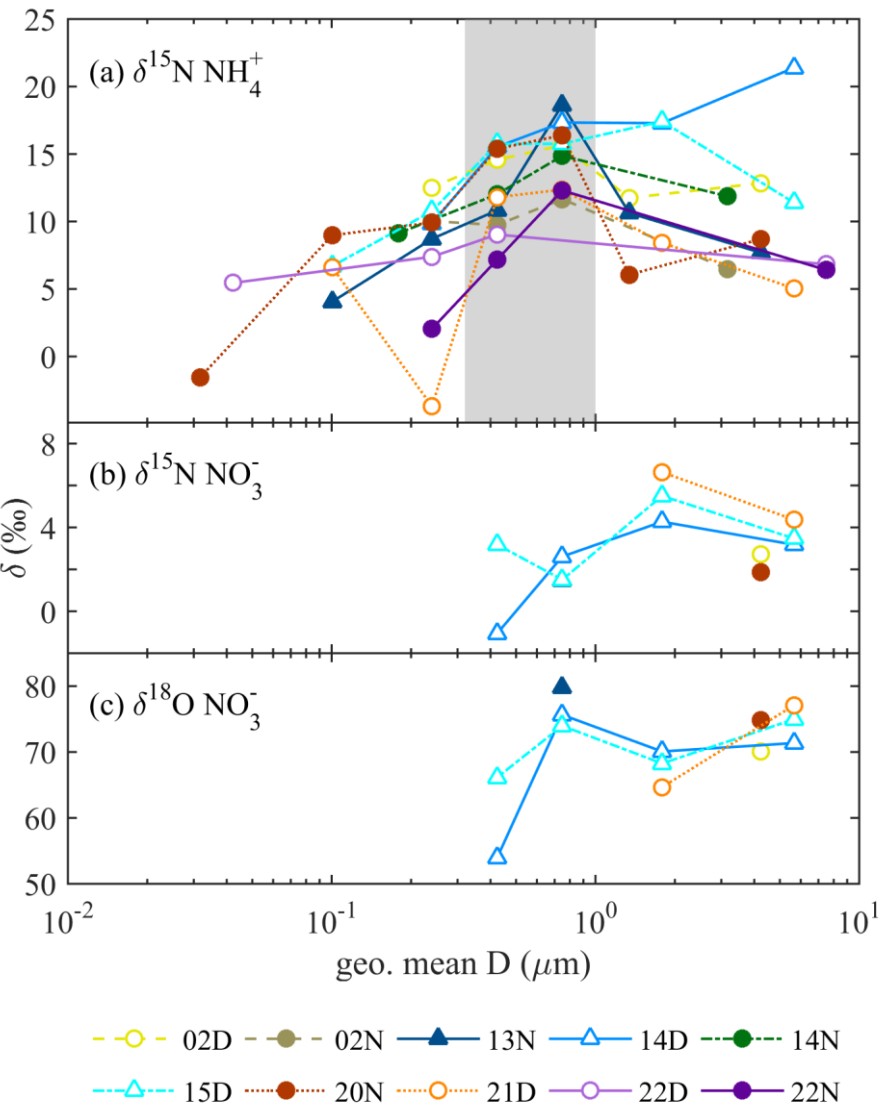

**Figure 3: The isotope values as a function of collected aerosol geometric mean diameter (D) (a) $\delta^{15}N$ $NH_4^+$, (b) $\delta^{15}N$ $NO_3^-$, and (c) $\delta^{18}O$ $NO_3^-$. The symbol is hollow for daytime, filled for nighttime, and triangle for foggy events, respectively.**

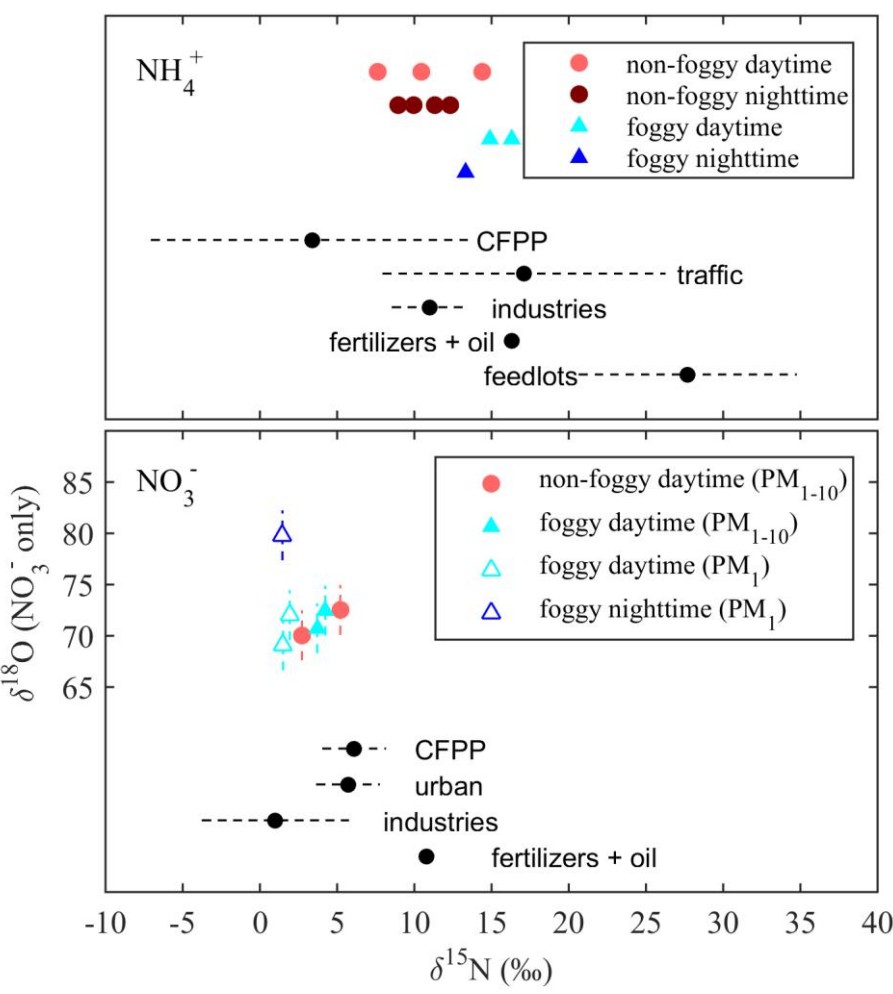

**Figure 4: Comparison between the period mass-averaged isotope values ($\delta^{15}$N and $\delta^{18}$O) and the mean $\delta^{15}$N value (black dots) by Savard et al. (2017) for different sources. The dashed lines are the standard deviation of the measurements. The batch SD of international standards' duplicates was 0.04 - 0.11‰ for $\delta^{15}$N (not observable in this figure), and 2.20 - 2.33‰ for $\delta^{18}$O as shown at each data point.**