# Peer review of "Source Apportionment and Evolution of N-containing Aerosols at a Rural Cloud Forest in Taiwan by Isotope Analysis"

_Atmospheric Chemistry and Physics, 2022_

## Author Comment (AC1)

We would like to thank Dr. Andrius Garbaras for his comments that improved the clarity and readability of the manuscript. Our point-by-point responses are found below in blue ink. The revised content is in yellow highlighted.

1. I would like to see more details on the measurement of the isotope ratio in the samples itself. This is actually a research that requires a lot of mastery because of the small amounts of analyte encountered. I would like the authors to provide more details in the supplementary material: what was the linearity of the spectrometer, what smallest samples did the authors measure with sufficient accuracy, or was the linearity tested with international standards of various sizes? All of these details will be useful to readers who apply similar analysis in the future.

A: The following paragraph is added to the supplementary material to provide the required information for readers:

All $\delta^{15}N$ and $\delta^{18}O$ have been analyzed at Ren's lab at Department of Geosciences, National Taiwan University, using 'denitrifier method'. We use denitrifying bacteria strains *Pseudomonas aureofaciens* for $\delta^{15}N$ and $\delta^{18}O$ analyses on nitrate samples, and *Pseudomonas chlororaphis* for $\delta^{15}N$ analyses on total N samples after oxidizing reduced N forms to nitrate. The analytical errors for $\delta^{15}N$ and $\delta^{18}O$ of nitrate samples are generally smaller than 0.1‰ using the 'denitrifier method' for samples containing 5 nmol N or more (Fig. S9). The errors become slightly bigger with smaller samples, e.g., at 2 nmol N. As a result, we have only analyzed samples with 5 nmol or more N. The linearity on the current setup is within 0.2‰ between 5nmol and 20 nmol of N. But this does not affect our analyses, since we can correct for the linearity effect by analyzing samples and standards with constant N levels. Prior to isotopic analyses, we measure N concentration in each sample, so we could estimate the volume of samples needed to yield constant N amount (i.e., 5 nmol N). In addition, these samples are analyzed with standards at the same N level, such that any linearity effect will be sufficiently corrected. In addition, samples with very low nitrate concentration (less than 0.5 µmol/L in the dissolved solution) have greater errors for $\delta^{18}O$ analyses due to oxygen exchange effect with water during nitrate conversion. As a result, we only analyze samples that can yield greater than 1 µmol/L nitrate in the final dissolved solution. Samples or sample sizes will be binned if there is not sufficient N on each filter. Furthermore, we analyze samples with standards of similar concentration range. For example, samples with 7 µmol/L nitrate are analyzed with standards of 5 and 10 µmol/L nitrate, so the data correction using the nitrate standards also excludes uncertainties with different nitrate concentrations among samples. The above procedures are applied to all samples, which intend to address most if not all the uncertainties associated with isotopic analyses on nitrate samples. For total dissolved nitrogen, we use potassium persulfate reagent (3 g of Persulfate potassium and 5 g of Sodium hydroxide in 100 ml of Milli-Q water) to oxidize reduced N to nitrate prior to isotopic analyses. The main source of uncertainty in this oxidation step is associated with the blank of potassium persulfate reagent. We account for this uncertainty by using purified potassium persulfate after 3 times of recrystallization, which typically yield blank size of 0.4 µmol/L N, and account for 6% of the total oxidized sample on average. In addition, we also process 5 blanks and 3 to 4 oxidation standards using international standards USGS 40 ($\delta^{15}N= -4.52$ ‰) along

with each run (typically containing less than 30 samples). The oxidation standards cover the range of blank/sample ratio in the samples, so we could also correct for blanks. Although we did not perform duplicates for the oxidation plus isotopic analyses on our samples, the 1 standard deviation for oxidation standards is less than 0.21 ‰, which represents the uncertainty for isotopic analyses for oxidized TN samples.

[Figure]

Figure S9. Measured $\delta^{15}N$ of IAEA N3 (open black circles) and USGS34 (closed red circles) at different nitrogen levels. The black and red lines indicate the true values of the two standards. 1std of $\delta^{15}N$ at each nitrogen level is ~0.1 ‰. The changes in the measured $\delta^{15}N$ at different nitrogen levels reflect the current linearity of the system, which would be corrected with standards.

Some specific comments:

2. Line 75 It's not clear where samples were collected. It's written that in Xitou experimental forest, but is not clear the location is up in the hill or in valley.

A: Xitou experimental forest is located in a valley as shown in Figure R1. The content is revised as: "A field campaign was conducted over Xitou experimental forest (23°40'12'' N, 120°47'54'' E, 1,179 m a.s.l.) in a valley from 1st to 24th December 2018 to investigate the interaction between air quality, local circulation, and human activities in central Taiwan."

[Figure]

Figure R1. The topographic map nearby Xitou Experimental Forest (adapted from Google Maps). The red circle is the sampling location.

3.      Line 105. There is no description how BC was measured with FTIR-ATR analysis. Does it is comparable with the measurements with other BC techniques, for example aethalometer?

A:   Because BC absorbs broad radiation, the absorbance of BC was determined by the average absolute absorbance in the region of $3950 \pm 5$ cm$^{-1}$ where the interference by other chemical species is negligible, as shown in Fig. S1 (the whole baseline shifted up). The calibration of BC absorbance at $3950 \pm 5$ cm$^{-1}$ was performed in the earlier study (Huang, 2016), with the elemental carbon concentration determined using a DRI2001A carbonaceous aerosol analyzer, following the IMPROVE thermo-optical reflectance (TOR) protocol (Chow et al., 2001), as detailed in Chou et al. (2010). The BC measurement is clarified with the following statement added to the end of section 2.2: "As to black carbon (BC) concentration, the absolute absorbance at $3950 \pm 5$ cm$^{-1}$ is applied to quantify the BC concentration based on the calibration done by Huang (2016) with the elemental carbon concentration determined using a DRI2001A carbonaceous aerosol analyzer, following the IMPROVE thermo-optical reflectance (TOR) protocol (Chow et al., 2001), as detailed in Chou et al. (2010).".

References:

Huang, R.-T.: A study of aerosol hygroscopicity in Kinmen, Graduate Institute of Atmospheric Sciences, National Taiwan University, Taipei, Taiwan, 10.6342/NTU201603559, 2016.

Chow, J. C., Watson, J. G., Crow, D., Lowenthal, D. H., and Merrifield, T.: Comparison of IMPROVE and NIOSH Carbon Measurements, Aerosol Science and Technology, 34, 23-34, 10.1080/02786820119073, 2001.

Chou, C. C.-K., Lee, C. T., Cheng, M. T., Yuan, C. S., Chen, S. J., Wu, Y. L., Hsu, W. C., Lung, S. C., Hsu, S. C., Lin, C. Y., and Liu, S. C.: Seasonal variation and spatial distribution of carbonaceous aerosols in Taiwan, Atmospheric Chemistry and Physics, 10, 9563-9578, 10.5194/acp-10-9563-2010, 2010.

4.      Line 135. What stands for letter p in "p-NO3 -=…"

A:   The letter p stands for particulate phase. However, in this study, only particulate $NO_3^-$ is discussed. The letter "p-" is deleted in the revision.

5.      Line 170. Fig. 2(c) and 2(d). NH4+ is not in the Fig. 2(d).

A:   The reviewer is correct. Figures 2(c) and 2(d) are for $NH_4^+$ and $NO_3^-$, respectively. The content in Line 170 is revised as " The mass distribution seems to shift to a larger size bin (0.56-1.8μm) for $NH_4^+$ as shown in Fig. 2(c), while $NO_3^-$ in Fig. 2(d) has a significantly high concentration for the 0.56-1.8μm size bin during the foggy period."

6.      Line 170. It's not clear boundary level effect. Does it mean that the boundary level is always above the sampling station?

A:   During daytime, the land was heated by solar radiation, causing boundary layer height to rise to a higher altitude (~1-2 km). The daytime boundary layer height mostly above the sampling site. The foggy period is likely associated with a stronger boundary layer inversion, which has a lower boundary height but is still above the sampling site. However, the sample site is likely below the nighttime boundary layer height as it was estimated to be less than 600 m a.s.l. based on the radio-sounding measurements at the foot of the hill nearby. We add the following sentence to the end of section 3.1.1 for clarification: "The sampling site is mostly below the boundary layer height during daytime and above the boundary layer height during nighttime."

7.      Line 190. I look at Fig. 3a and I see on average lower $\delta^{15}N$ values in submicron range comparing to bigger particles. Authors say that the "trend of a higher $NH_4^+$ $\delta^{15}N$ in submicron aerosol was also observed in Beijing". I do not understand how Authors compare different size bins.

A: The statement "a higher $NH_4^+$ $\delta^{15}N$ in submicron aerosol" didn't provide an accurate description. For a given collection period data, $\delta^{15}N$ values generally show an increasing trend first and then a decreasing trend with particle size. The maximum $\delta^{15}N$ happens around the 0.56-1 µm size bin for most non-foggy daytime. The sentence is revised as "The increasing and then decreasing trend of $NH_4^+$ $\delta^{15}N$ with aerosol size was also observed in Beijing…" to provide a more accurate illustration.

8.      Line 195. What is mean "daytime fractionation"?

A: We use "daytime fractionation" to describe the fractionation that happened during daytime. To avoid confusion, the content is modified as "As the mountain wind dominates after sunset, available $NH_3$ might be attributed to the daytime residual (having lower $\delta^{15}N$ due to the fractionation that happened during daytime) or the local biogenic sources having a lower $\delta^{15}N$."

9.      Line 200. PM1-10 was higher … similar to 0.32-1 µm aerosol. So no difference in all size bins, as almost the whole range fall in the 0.32 – 10 µm. This kind of differentiation seems artificial.

A: Yes, the $NH_4^+$ $\delta^{15}N$ of foggy daytime is relatively flat at a diameter larger than 0.56 µm. However, $\delta^{15}N$ for $PM_{1-10}$-$NH_4^+$ at foggy daytime is higher than that at non-foggy conditions. It might be attributed to the growth of part of 0.56-1 µm aerosols under high RH. To improve the clarity, the content in this paragraph is revised as follows: " Fog varies the composition mass distribution among different size bins and can affect the isotopic ratio. Under foggy daytime conditions, the $\delta^{15}N$ value of larger size aerosols ($PM_{1-10}$-$NH_4^+$) was more like to be the extension of 0.56-1 µm with a value up to 21.39‰, higher than that of non-foggy days. As stated in section 3.1, high $NH_3$ concentration can promote the partition of $HNO_3$ during foggy conditions to enhance hygroscopicity. The observed flat trend of $\delta^{15}N$ at dimeter $\geq 0.56$ µm might result from the hygroscopic particle growth of $NH_4^+$ from the 0.56-1 µm size bin aerosols. "

10.  Fig. 2. The legend must be revised. I suggest adding a legend to the (b) and (d) for

clarification.

A:  Thanks for Dr. Garbaras' comment. We added a legend to Fig. 2(b) and (d) and adjusted the legend location for clarification. The updated figure is as follows:

[Figure]

---

## Author Comment (AC2)

**We would like to thank the anonymous reviewer for the comments that significantly improve the clarity and readability of the manuscript. Our point-by-point responses are found below in blue ink. The revised content is highlighted in yellow.**

The authors report results from an aerosol sampling campaign in a rural cloud forest during December 2018. Different size fractions were sampled on filters taken during daytime and night-time and during some days fog events impacted the aerosol composition. The most important measured aerosol components were ammonium, nitrate, sulphate, and black carbon. Ammonium and nitrate were also analysed for stable isotopes no nitrogen and oxygen.

The study nicely show local dynamics of aerosols and their partitioning into different size fractions. Differences in stable oxygen isotopes of nitrate during foggy conditions revealed a possible oxidation pathway involving peroxyl radicals.

1.      My major concern is the performed source apportionment using the stable nitrogen isotopes and a mixing model (MixSIAR). Many aspects of the procedure are insufficiently described (e.g. what is posterior in this context, and how should probabilities interpreted). Table 4 seems to list the results of the source apportionment. I see mostly values around 20 with standard deviations around 15. A threshold of 20 is applied, but the choice of this value is not motivated. Overall, most values do not seem to be significantly different. I fail to see how any conclusions can be drawn from this model. Therefore, I suggest to remove this part.

A: The MixSIAR is a Bayesian mixing model to infer the probable sources of a mixture using given prior information. In this study, the mean values and standard deviation of stable isotope from different sources in a previous study was applied as the prior data and assumed to have a Gaussian distribution. After applying the Bayes' theorem, the posterior probability is the conditional probability based on these observation data. The following description in Lines 146-150 is revised as follows for clarification: "MixSIAR is a statistical model applying Bayesian Inference to infer the posterior probability of mixture sources by analyzing its tracer composition, such as stable isotope or fatty acids (Stock et al., 2018). The studied tracers are assumed to transfer from sources to the mixture through a conserved mixing process integrating the observed variability. In this study, the observed mass-weighted $\delta^{15}N$ of $NH_4^+$ and $NO_3^-$ for each sampling period was used as prior information of the mixture."

        The similar isotope values for some applied source data (i.e., traffic, industries and fertilizers for $NH_4^+$ $\delta^{15}N$, and CFPP and urban for $NO_3^-$ $\delta^{15}N$) can lead to comparable posterior probabilities. However, the results can differentiate the sources with significantly different isotopes, such as relatively lower probabilities of feedlots and traffic in $NH_4^+$ $\delta^{15}N$, and fertilizers in $NO_3^-$ $\delta^{15}N$. With the source and sample variability, the results of MixSIAR provide broader probabilities for source contribution, which might reflect the uncertainty of the ambient conditions. However, the possible differentiation among the similar $\delta^{15}N$ sources might require the integration of the back trajectory and model simulation with the known emission sources. In the content, the following information is added in Lines 282-

286 (section 3.3) to address this issue; "The similar posterior probabilities among some sources are due to the comparable source isotope values as stated above. However, with the source and sample variability, the results of MixSIAR provide a broader probability for source contribution and reflect the uncertainty of the ambient conditions simply using the mixing rule. The possible differentiation among the similar $\delta^{15}N$ sources might require the integration of the back trajectory and chemical transport model simulation with the known emission sources."

Minor comments:

2.      Language needs to be improved. Several issues… already in the first sentence of the abstract (aerosol components NOT compositions).

A: Thanks for the reviewer's comment. We went through the content to correct the word and grammar for clarification. Some examples are shown as follows:

Line 9. "Ammonium and nitrate are major N-containing aerosol components."

Line 36. "Ammonium and nitrate are the primary N-containing cation and anion species, respectively, …"

3.      Figure 1: extend the figure to also indicate how daytime and night-time chemistry results in different stable isotope composition. A good description is given in the supplement. Maybe some of this can be incorporated in Fig 1.

A: Fig. 1 and the figure caption are revised with the isotope values from fresh and aged gas precursors via different chemical pathways as follows:

[Figure]

Figure 1. The formation pathway of nitric acid to form aerosol nitrate during daytime (orange color) and nighttime (blue color) with the predicted $\delta^{18}O$ range of $NO_3^-$ based on

(a) freshly emitted NO and (b) NO cycled from $NO_2$, fully reacted with $O_3$ (detail can be found in Figures S6 and S7).

4. L132/133: I am not sure if organic nitrogen can be neglected. There are several papers out reporting organic nitrates and other organic nitrogen compounds in aerosols. The authors should at least discuss how their results would change of there are significant fractions of other nitrogen compounds.

A: The presence of organic nitrogen in aerosols is undeniable. However, the water-soluble reduced nitrogen, e.g., ammonium and organic nitrogen, can be estimated as the difference between total nitrogen and nitrate as WS(TN-NN). WS(TN-NN) shows a good correlation (slope is close to 1 with a small interception as shown in Figure S4) with the estimated ammonium determined using FT-IR. The result suggests that ammonium is the significant component of the reduced nitrogen for this studied case. However, the presence of organic nitrogen might lead to some deviation of the determined $\delta^{15}N$ $NH_4^+$. Organic nitrogen might be related to $NO_x$ and was reported a lower $\delta^{15}N$ than nitrate (Wu et al. 2021), less than -5‰. If organic nitrogen with a lower $\delta^{15}N$ than nitrate is taken into account, we can expect a slightly higher $\delta^{15}N$ $NH_4^+$ than the current reported values. We added the following sentence to Lines 141-143 to address this issue, "If organic nitrogen is considered, a slightly higher $\delta^{15}N$ of $NH_4^+$ than the current reported values can be expected because organic nitrogen might be related to $NO_x$ and was reported a lower $\delta^{15}N$ ($\leq$ -5‰) than nitrate (Wu et al., 2021)."

Wu, L., Yue, S., Shi, Z. *et al.* Source forensics of inorganic and organic nitrogen using $\delta^{15}N$ for tropospheric aerosols over Mt. Tai. *npj Clim Atmos Sci* **4,** 8 (2021). https://doi.org/10.1038/s41612-021-00163-0

5. L214-216: This sentence has language issues. The argumentation does not seem to be logical.

A: The sentence is revised in Lines 225-228 to clarify the argument as follows: "As stated in section 3.1.2., nitrate significantly contributes to the submicrometer particles during foggy daytime in addition to the usual peak over the supermicrometer particles for all conditions (Fig. 2). The nitrate can be divided into two groups, $PM_{1-10}\text{-}NO_3^-$ for particle size in the range of 1 to 10 μm and $PM_1\text{-}NO_3^-$ for particle diameter less than 1 μm, for further discussion."

6. L228-230: Was there any evidence for agricultural activity during that period? What was different compared to other periods?

A: Since the sampling site was in a nursery of the experimental forest, some agricultural activities happened during the observation period. We recorded that fertilizers were applied on the field on December 18th, and the scheduled mowing activities nearby the sample collection site were on the daytime of December 20th and 21st. The sentence is revised as

" The sample of 21D is a special case with higher $\delta^{15}N$ values. It might result from the recorded agricultural activities nearby,…".

7.      L281-282: "The posterior probability of PM1 and PM1-10 nitrate sources has difference slightly:" This seems to be a mixture of poor English with lab/model-slang.

A: The sentences are revised in Lines 296-300 as follows: "The difference in posterior probability between PM$_1$ and PM$_{1-10}$ nitrate sources is not significant: the PM$_{1-10}$ NO$_3^-$ was more likely from CFPP, industries, or urban sources, while industries had the majority of PM$_1$-NO$_3^-$ formation. However, the inferred source difference might suggest that the coarse mode aerosols came from the coastal sea salt particles mixing with the emission of coal-fired power plants or the Taichung-Changhua metropolitan during the inland transport."